# Multichannel transcranial direct current stimulation over the left dorsolateral prefrontal cortex may modulate the induction of secondary hyperalgesia, a double-blinded cross-over study in healthy volunteers

Arnaud Steyaert[1,2]*, Cédric Lenoir[1], Patricia Lavand'homme[1,2], Emanuel N. van den Broeke[1], André Mouraux[1]

1 Institute of Neuroscience (IONS), Université catholique de Louvain (UCLouvain), Brussels, Belgium,
2 Departement of Anaesthesiology, Cliniques Universitaires Saint-Luc, Brussels, Belgium

* arnaud.steyaert@uclouvain.be

## Abstract

### Background

Central sensitization is thought to play a critical role in the development of chronic pain, and secondary mechanical hyperalgesia is considered one of its hall-mark features. Consequently, interventions capable of modulating its development could have important therapeutic value. Non-invasive neuromodulation of the left dorsolateral prefrontal cortex (DLPFC) has shown potential to reduce pain, both in healthy volunteers and in patients. Whether it can modulate the induction of central sensitization, however, is less well known.

### Objective

To determine whether multifocal transcranial direct current stimulation (tDCS) targeting the left DLPFC affects the development of secondary mechanical hyperalgesia.

### Methods

In this within-subjects, cross-over, double-blinded study, eighteen healthy volunteers participated in three experimental sessions. After 20 minutes of either anodal, cathodal, or sham multichannel tDCS over the left DLPFC, secondary mechanical hyperalgesia was induced using high-frequency electrical stimulation (HFS) of the volar forearm. We assessed intensity of perception to 128 mN mechanical pinprick stimuli at baseline and up to 240 minutes after HFS. We also mapped the area of mechanical hyperalgesia.

### Results

HFS resulted in a robust and unilateral increase in the intensity of perception to mechanical pinprick stimuli at the HFS arm, which was not different between tDCS stimulation

**Data Availability Statement:** The data are held in a public repository, https://dataverse.uclouvain.be.

**Funding:** AS is supported for this research by a grant from the Fonds de Recherche Clinique (Cliniques Universitaires Saint-Luc, Belgium) and from the BESARPP (Belgian Society for Anaesthesiology and Reanimation). The funders had no role in study design, data collection and analysis, decision to publish, or preparation of the manuscript.

conditions. However, the area of hyperalgesia was reduced after anodal tDCS compared to sham.

## Conclusion

Anodal tDCS over the left DLPFC modestly modulates the size of the HFS-induced area of secondary mechanical hyperalgesia, suggesting that non-invasive neuromodulation targeting the left DLPFC may be a potential intervention to limit the development of central sensitization.

## Introduction

Non-invasive brain modulation techniques such as transcranial direct current stimulation (tDCS) and repetitive transcranial magnetic stimulation (rTMS) are emerging as complementary pain treatment options, both for postoperative [1,2] and chronic pain [3]. Compared to rTMS, tDCS has the advantage of being cheap, easy to administer, and well tolerated, making it an ideal candidate for clinical use [4]. Conventional tDCS, delivered through a single pair of large surface electrodes, induces a rather diffuse electric field (*E*-field). In contrast, more recently developed multichannel tDCS systems can increase the spatial focality of stimulation by weighting the currents delivered through multiple electrodes placed on the scalp to maximize the *E*-field at the desired target [5,6].

A potential target to reduce pain using focused brain modulation techniques is the dorsolateral prefrontal cortex (DLPFC). The DLPFC is a functionally and structurally heterogeneous brain region, which plays a vital role in emotion, cognition, behaviour, and pain processing [7]. In healthy volunteers, non-invasive neuromodulation targeting the DLPFC decreases pain sensitivity [8–11] and both intensity and painful body area in an experimental model of sustained muscle pain [12,13]. In patients suffering from acute pain after surgery, modulation of the left DLPFC using anodal tDCS or rTMS reduces analgesic consumption [14–16] and pain intensity [15,17]. Moreover, a recent Cochrane review concluded that there is evidence, albeit of very low quality and only from four studies, that rTMS over the DLPFC may reduce pain intensity and improve the quality of life of patients with chronic pain [18].

Central sensitization–defined by the International Association for the Study of Pain as the increased responsiveness of nociceptive neurons in the central nervous system to their normal or subthreshold afferent input–is thought to play a critical role in the development of persistent post-surgical pain and other chronic pain conditions [19]. Secondary mechanical hyperalgesia–i.e., the increased sensitivity to mechanical nociceptive stimuli that develops beyond the area of tissue injury and tissue inflammation–is considered a hallmark feature of central sensitization [20]. In patients that have undergone surgery, the size of the area of secondary hyperalgesia around the surgical wound is associated with the risk of persistent post-surgical pain, sparking interest in interventions capable of modulating the post-operative development of secondary hyperalgesia [21–23]. In healthy human volunteers, a state of increased sensitivity to mechanical nociceptive stimuli (i.e. secondary hyperalgesia) lasting up to several hours can be induced experimentally using techniques which produce sustained nociceptive input such as high-frequency electrical stimulation (HFS) of the skin, via an electrode designed to preferentially activate superficial cutaneous nociceptors [24].

Secondary hyperalgesia has been explained as resulting from an enhanced synaptic transmission of nociceptive input at the level of the spinal cord [25]. Importantly, nociceptive

circuits at spinal level can also be facilitated or inhibited by descending modulation pathways originating from supra-spinal structures [26]. Recent animal work by Tan *et al.* demonstrated that the midcingulate division of the cingulate cortex (MCC) is crucial in regulating mechanical hypersensitivity behaviour after intraplantar capsaicin injection in mice [27]. While it is difficult to target this deep brain structure using current non-invasive neuromodulation techniques, an indirect modulation of the MCC could be achieved by targeting the left DLPFC. Indeed, Stagg *et al.* [28] showed that anodal tDCS over the left DLPFC induces widespread changes in brain perfusion, including in the MCC [25]. Moreover, rTMS targeting the left DLPFC has been shown to modulate hyperalgesia in other experimental models of pain [12,13].

With the current study, we aimed to determine whether non-invasive neuromodulation of the left DLPFC can influence the development of secondary mechanical hyperalgesia. To answer this question, we conducted a randomized double-blinded study in healthy human volunteers testing the effect of a single session of multichannel tDCS targeting the left DLPFC on the intensity and spatial extent of the secondary mechanical hyperalgesia induced by HFS applied onto the volar forearm skin.

## Material and methods

### Participants

Healthy volunteers were recruited through online advertising. Exclusion criteria were known medical conditions (e.g., diabetes, neuropathy, psychiatric disorders, seizure, migraine, presence of a pacemaker or other implanted medical devices) and use of any medication, except for contraception. We planned to collect data from 18 participants. This sample size was based on the sample sizes of comparable studies assessing effects of tDCS on nociceptive processing in healthy volunteers (see [29] for a review) and on the necessity to choose a multiple of six to be able to counterbalance the order of the three sessions across participants. At the time of enrolment, all participants received written information about the experiment and gave their informed consent. The study was approved by an independent ethical committee (B403201316436, Commission d'Éthique Biomédicale Hospitalo-Facultaire, Université Catholique de Louvain). All experimental procedures were conducted in accordance to the latest revision of the Declaration of Helsinki.

### Experimental design

In this within-subject study, each volunteer participated in three experimental sessions, separated by a minimum of two weeks to avoid any after-effects of HFS or tDCS. All three sessions were identical except for the tDCS protocol (anodal, cathodal, or sham stimulation). The order of the tDCS sessions was assigned randomly and counterbalanced across the participants, through a randomisation table generated with a custom MATLAB script before the start of the study. A single investigator conducted all the acquisitions. Both this investigator and the participant were blinded to the stimulation parameters.

Each session started with a baseline sensory testing, immediately followed by a 20-minute multifocal tDCS session. Twenty-five minutes after the end of tDCS, a second sensory testing was performed. Then, HFS was applied to the volar forearm, and three additional instances of sensory testing were performed 20, 40, and 240 minutes after HFS. The decision to apply HFS approximately 30 minutes after the end of the tDCS session was justified by evidence indicating that anodal tDCS targeting the left DLPFC increased M1 excitability and pain pressure threshold may build up over time for at least such a delay [30].

Before HFS, the sensory testing consisted of measuring the detection threshold to a single electrical pulse delivered using the HFS electrode, and the intensity of perception to 128 mN

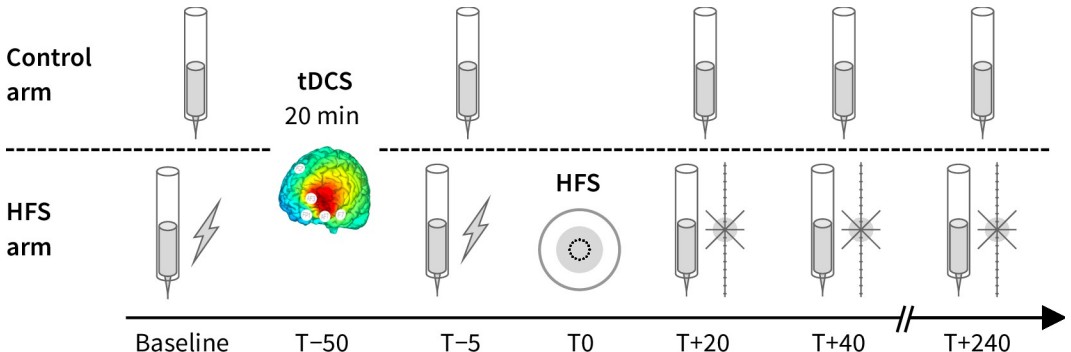

**Fig 1. Timeline of each experimental session.** All sessions were identical, except for the tDCS protocol (anodal, cathodal or sham stimulation). On both arms (HFS arm and control arm), we assessed the intensity of perception to 128 mN mechanical pinprick stimuli at baseline, 5 minutes before applying HFS (T-5), and 20, 40, and 240 minutes after HFS application (T+20, T+40, T+240). The detection threshold to single electrical pulses delivered through the HFS electrode was assessed at the HFS arm, at baseline and T-5. The extent of the area of hyperalgesia was mapped at the HFS arm at T+20, T+40 and T+240.

mechanical pinprick stimuli at both forearms. After HFS, the sensory testing consisted of measuring the intensity of perception to 128 mN mechanical pinprick stimuli at both forearms and the extent of the area of increased sensitivity to mechanical pinprick stimuli at the HFS-treated forearm. Fig 1 details the timeline of each experimental session.

## Transcranial direct current stimulation

Multichannel tDCS of the left DLPFC was delivered by an 8-channel Starstim current generator (Neuroelectrics, Barcelona, Spain). We used a custom 5-electrode montage, based on the modelling work by Ruffini *et al.* and recommended by the manufacturer for use with the Starstim current generator [6]. The overall aim of such multichannel montages is to maximize the normal $E$-field (En) on target, while minimizing the En elsewhere. After defining the target area (left DLPFC), the En in the target area (0.25 V/m), the size and number of electrodes (3.14 cm$^2$ and 5), and the maximum current per electrode (2 mA), the optimal multichannel montage was determined by minimizing the least-square difference between the resulting $E$-field and a weighted target map of the En. The five 3.14 cm$^2$ electrodes (Pistim, Neuroelectrics, Barcelona, Spain) were positioned using a neoprene head cap at locations AF3, AF7, FP1, F7 and FZ according to the standard 10–20 system. The stimulation consisted in a 2-mA current weighted as follows across the different electrodes (AF3: ±1168 μA, AF7: ±1830 μ A, FP1: ±2000 μ A, F7: ±749 μ A, FZ: ±249 μ A). tDCS polarity was defined by the polarity of the AF3 and AF7 electrodes. As shown in Fig 2, this setup was expected to maximize the $E$-field over the left DLPFC. During the anodal and cathodal sessions, stimulation started with a 30-second ramp-up time, after which the 2-mA current was maintained for 20 minutes. At the end of stimulation, the current was ramped down in 30 seconds. During the sham session, the montage was identical, and the stimulation was always anodal, but consisted only of a 30-second ramp-up followed by a 30-second ramp-down at the start of the stimulation and 20 minutes later. The order of the sessions for each participant was preloaded in the NICLab software (version 1.05, Neuroelectrics, Barcelona), which was used in "blind mode". At the end of each session, participants were asked to guess if they had received active or sham stimulation.

## High-frequency electrical stimulation

HFS consisted of 5 trains of 100 square-wave electrical pulses (pulse-width 2 ms), delivered at a frequency of 100 Hz. Each train (lasting 1 s) was separated by a 10-second inter-train

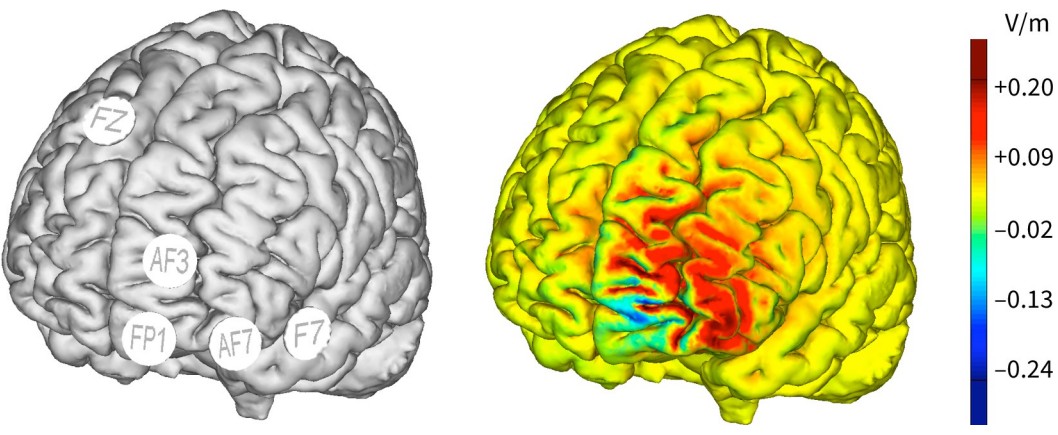

**Fig 2. Setup of tDCS stimulation in the NICLab software.** Left: Electrode position for multichannel tDCS targeting the left DLPFC. Polarity was defined by the polarity of the AF3 and AF7 electrodes. Right: Modeling of the distribution of the normal component of the E-field during multichannel anodal tDCS stimulation (AF3: 1168 μA, AF7: 1830 μ A, FP1: −2000 μ A, F7: −749 μ A, FZ: −249 μ A).

interval. The electric pulses were generated by a constant current electrical stimulator (DS7, Digitimer, UK), controlled by a programmable impulse generator (Master 8, AMPI, Jerusalem, Israel) and delivered to the skin via a custom electrode designed and built at the Centre for Sensory-Motor Interaction (Aalborg University, Denmark). The cathode consists of 16 blunt stainless-steel pins with a diameter of 0.2 mm, protruding 1 mm from the base and positioned in a 10 mm diameter circle. The anode is a stainless-steel ring with an inner diameter of 22 mm and an outer diameter of 40 mm (Fig 3, panel A) [24,31].

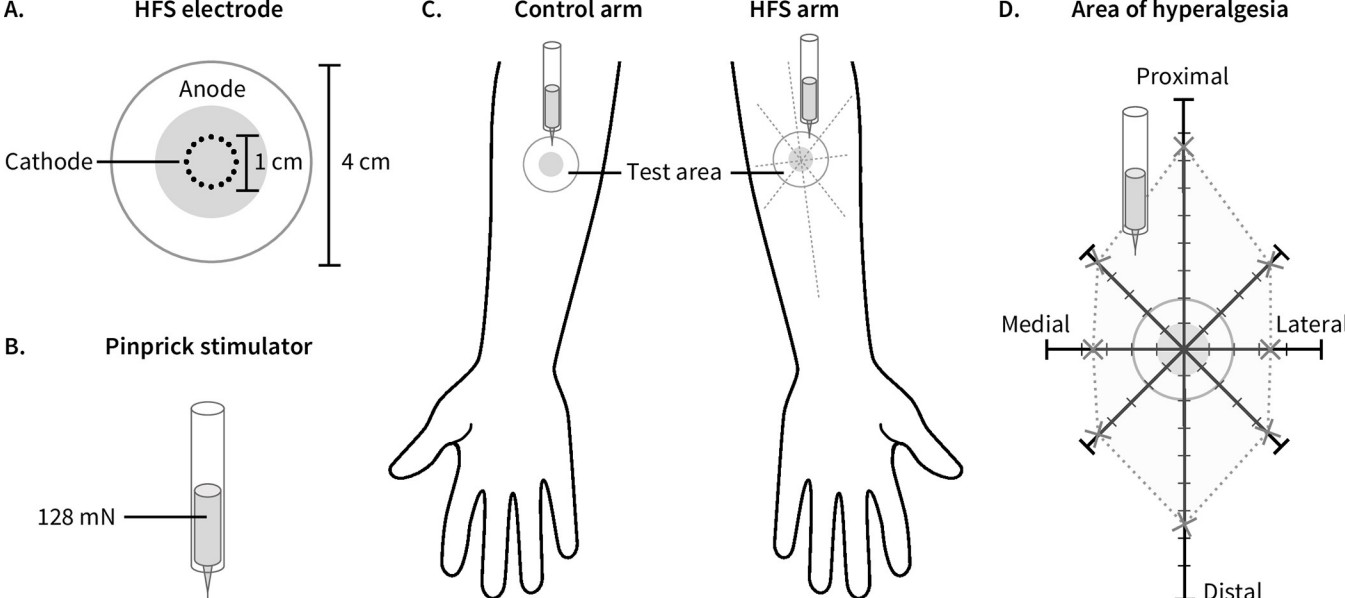

**Fig 3. Experimental setup.** (A) Electrode used to deliver HFS with its 16 blunt stainless-steel pins placed in a 10-mm diameter circle (cathode), surrounded by the concentrically located stainless steel anode. (B) We used a 128 mN mechanical pinprick stimulator for all measures of intensity of perception to mechanical stimuli. (C) We applied high frequency electrical stimulation of the skin (HFS) to the dominant volar forearm. We assessed the intensity of perception to mechanical stimuli in the area surrounding the area where HFS was applied, as well as in the same skin area on the contralateral (control) arm. (D) We mapped the size of the area of increased pinprick sensitivity by applying mechanical pinprick stimuli every centimetre along eight different radial axes, from the periphery towards the centre of the electrode.

The detection threshold to a single electrical pulse delivered through the HFS electrode was assessed twice, before and after tDCS, using the method of limits. The intensity of the first stimulus was set to 1.0 mA, after which we decreased the intensity of subsequent stimuli until the participant could no longer detect the stimulation. The threshold was confirmed by stimulating at least twice with intensities just below and above the threshold. The intensity of HFS was set to 20 times the electrical detection threshold assessed after tDCS.

## Mechanical stimulation

We used a calibrated 128 mN mechanical pinprick stimulator (MRC Systems GmbH, Heidelberg, Germany) to assess the perceived intensity elicited by mechanical pinprick stimulation of the skin (Fig 3, panel B). At each time-point, the stimuli were applied three times, perpendicular to the skin, within a circular area (radius 20 mm) surrounding the central point at which HFS was applied and within an identical area on the control arm, in a pseudo-randomised order (Fig 3, panel C). To prevent sensitization of the skin due to repeated pinprick stimulation, we never stimulated the exact same location twice. After each stimulus, participants rated the perceived intensity using a numerical rating scale (NRS) ranging from 0 (no detection) to 100 (maximal imaginable pain). A rating of 50 represented the transition between non-painful and painful sensations [32].

In addition to the intensity of perception, we also mapped the size of the area of increased pinprick sensitivity by applying mechanical pinprick stimuli every centimetre along eight different radial axes, from the periphery towards the centre of the electrode (Fig 3, panel D). With eyes closed, participants indicated the moment at which the perception of the stimulus increased markedly. The stimulus was then re-applied one centimetre distal to this point and proceeded towards the centre until the participant again reported an increase in perception intensity. We marked this point with a felt marker and measured the distance from the centre of the electrode. We calculated the surface of the area of increased pinprick sensitivity by fitting the polygon connecting the eight borders of the area with the best-fit linear segments.

## Statistical analysis

Statistical analyses were performed using *JMP Pro* 15.0 (SAS Institute Inc., Cary, NC, USA) and *Jamovi* 1.2 (The Jamovi project, https://www.jamovi.org). Descriptive statistics were used to present participant demographic characteristics.

To assess blinding, Cohen's kappa measure of agreement (κ) was used to test, for each session, whether participants correctly judged their stimulation condition (active *vs.* sham). Agreement is usually considered poor if κ < 0.00, slight if 0.00 ≤ κ ≤ 0.20, fair if 0.21 ≤ κ ≤ 0.40, moderate if 0.41 ≤ κ ≤ 0.60, substantial if 0.61 ≤ κ ≤ 0.80 and almost perfect if κ > 0.80 [33].

We assessed the distribution of the dependent variables by visually examining the histograms and Q-Q plots. All outcomes (electrical detection threshold, intensity of perception, and area of mechanical hyperalgesia) were highly skewed to the right. A square root transformation allowed us to obtain a secondary normal distribution.

To analyse the effects of tDCS on the *electrical detection threshold*, we fitted a linear mixed model on the square root of the electrical detection threshold with as fixed effects "time" (baseline and T–5), "stimulation" (anodal, cathodal, and sham), and their interactions. To analyse the effects of tDCS on the *intensity of perception*, we fitted a linear mixed model on the square root of the NRS scores with as fixed effects "arm" (HFS and control arm), "time" (baseline, T–5, 20, 40, and 240 minutes), "stimulation" (anodal, cathodal, and sham), and their interactions. To analyse the effects of tDCS on the *area of hyperalgesia*, we fitted a linear mixed model on

the square root of the area in which pinprick sensitivity was increased with as fixed effects "time" (20, 40, and 240 minutes), "stimulation" (anodal, cathodal, and sham), and their interactions. In all models, we included a random intercept for "participants" and "session". When *F*-values were statistically significant, we conducted post-hoc comparisons using the HSD Tukey method. We considered the level of significance at *p*-value ≤ 0.05.

## Results

### Participants

We recruited a total of 19 healthy volunteers: 12 females and 7 males. One female participant dropped out after the first session due to scheduling difficulties and was consequently replaced, resulting in a total of 18 participants with complete datasets for all three sessions. Demographics are reported in Table 1.

### Blinding and adverse effects

Overall, participants seemed adequately blinded to the polarity of stimulation. After session one, 12 participants correctly guessed their condition (66%, κ = 0.30, 95%IC –0.13 to 0.74). In session two this number dropped to 9 (50%, κ = –0.17, 95%IC –0.59 to 0.24) and in session three to 10 (55%, κ = –0.09, 95%IC –0.51 to 0.33).

Some participants experienced slight discomfort during tDCS, but we observed no serious adverse effects. Table 2 reports the incidence of adverse effects for each of the conditions.

### Electrical detection threshold before and after tDCS

Using a linear mixed model, we found no effect of time ($F_{(1, 83)} = 1.36$, *p* = 0.25), of stimulation ($F_{(2, 83)} = 1,74$, p = 0.18) or of their interaction ($F_{(2, 83)} = 1.37$, *p* = 0.26).

### Intensity of perception of pinprick stimuli before and after HFS

Fig 4 (left panel) shows the estimated marginal means of the square root of the perceived intensity to mechanical pinprick stimuli before and after HFS. Using a linear mixed model, we found a significant main effect of time ($F_{(4, 491)} = 40,4$, *p* < 0.0001), arm ($F_{(1, 491)} = 308.2$, *p* < 0.0001), and of the interaction of arm × time ($F_{(4, 491)} = 60.2$, *p* < 0.0001). The main effect of stimulation and the other interactions were not statistically significant. Post-hoc testing showed that the perceived intensity elicited by mechanical stimulation was similar for both arms at baseline but higher at the HFS arm at 20 (2.1, 95%IC [1.6, 2.5], t = 14.3, *p* < 0.0001), 40 (2.4, 95%IC [1.9, 2.9], t = 16.7, *p* < 0.0001) and 240 minutes (1.2, 95%IC [0.7, 1.6], t = 8.1, *p* < 0.0001) after stimulation, demonstrating an unilateral effect of HFS.

**Table 1. Demographics of the study participants.**

| | | |
|---|---|---|
| Age (years) | 23.5 | (4.0) |
| BMI (kg·m$^{-2}$) | 23.4 | (2.9) |
| Sex | | |
| Female | 11 | (61%) |
| Male | 7 | (39%) |
| Dominant arm | | |
| Right | 16 | (89%) |
| Left | 2 | (11%) |

Data are presented as mean (SD) or number (percentage).

**Table 2. Incidence of adverse events after stimulation.**

|  | Anodal |  | Cathodal |  | Sham |  |
|---|---|---|---|---|---|---|
| Headache | 4 | (22.2%) | 3 | (16.7%) | 4 | (22.2%) |
| Neck pain | 4 | (22.2%) | 2 | (11.1%) | 3 | (16.7%) |
| Blurred vision | 1 | (5.6%) | 1 | (5.6%) | 0 | (0.0%) |
| Scalp irritation | 3 | (16.7%) | 3 | (16.7%) | 0 | (0.0%) |
| Tingling | 3 | (16.7%) | 1 | (5.6%) | 1 | (5.6%) |
| Itching | 3 | (16.7%) | 1 | (5.6%) | 1 | (5.6%) |
| Burning sensation | 3 | (16.7%) | 3 | (16.7%) | 1 | (5.6%) |

Data are presented as number of occurrences (percentages).

### Extent of hyperalgesia area after HFS

Fig 4 (right panel) shows the estimated marginal means of the square root of the area of hyper-algesia 20, 40, and 240 minutes after HFS. Using a linear mixed model, we found a significant main effect of time (F (2, 134) = 26.4, $p < 0.0001$) and stimulation (F (2, 134) = 3.5, $p = 0.04$) on the length of the area of hyperalgesia, but no significant interaction. Post-hoc testing revealed that the area of hyperalgesia decreased significantly 240 minutes after HFS, compared to 20 (–21.9, 95%IC [–29.5, –14.4], t = –6.87, $p < 0.0001$) and 40 minutes (–17.5, 95%IC [–25.1, –9.9], t = –5.49, $p < 0.0001$). When averaged across time-points, the area of hyperalgesia was significantly smaller after anodal stimulation when compared to sham (–8.3, 95%IC [–15.8, –0.7], t = –2.59, $p = 0.03$) (Fig 5).

## Discussion

The results of this randomized, double-blinded, cross-over study comparing the effects of anodal, cathodal, and sham multifocal tDCS over the left DLPFC on HFS-induced secondary hyperalgesia in healthy volunteers show (1) no significant effect of tDCS on the HFS-induced increase in mechanical pinprick sensitivity and (2) a modest but nevertheless statistically significant effect of tDCS on the extent of the area of HFS-induced secondary hyperalgesia, which was reduced after anodal tDCS when compared to sham stimulation.

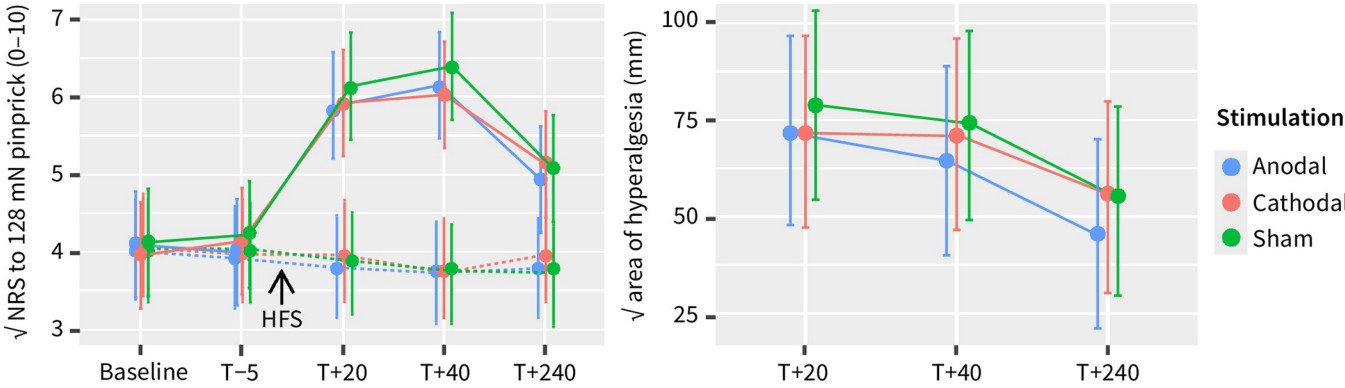

**Fig 4. Left: Comparison of the evolution over time of the intensity of perception to mechanical pinprick stimuli at the control (dashed lines) and the HFS (full lines) arm after anodal, cathodal, and sham multifocal tDCS.** Right: Comparison of the evolution over time of the square root of the area of secondary hyperalgesia (area in which the intensity of perception to mechanical pinprick stimuli was increased after HFS) after anodal, cathodal, and sham multifocal tDCS. In both panels, the data presented are estimated marginal means and 95% confidence intervals.

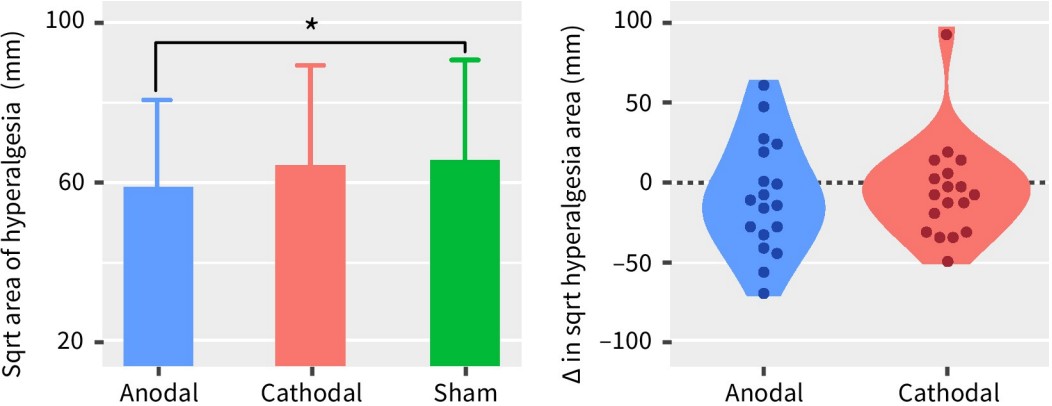

**Fig 5. Left: Comparison of the estimated marginal means of the extent of the hyperalgesia area after anodal, cathodal, or sham multifocal tDCS over the left DLPFC, averaged across timepoints, and expressed as the square root of the area of increased pinprick sensitivity (mm).** Data are presented as mean and 95% confidence interval. The asterisk indicates the significant difference between anodal and sham tDCS (Tukey HSD test, p = 0.02). Right: Differences in the area of hyperalgesia between sham and anodal or cathodal multifocal tDCS over the left DLPFC, averaged across time-points. Subject-level differences (anodal minus sham or cathodal minus sham) are represented by the dots. Negative values represent a smaller area of hyperalgesia for anodal or cathodal when compared to sham.

Non-invasive neuromodulation of the DLPFC, a brain region which plays a role in nociceptive processing, has been shown to decrease pain sensitivity in human volunteers [8,10,11], and has shown analgesic effects in patients [3,14,16]. The present study suggests that neuromodulation of the DLPFC may also influence the susceptibility to develop central sensitization and secondary hyperalgesia. In line with our results, functional neuroimaging studies have suggested that enhanced activity in the DLPFC is associated with enhanced sensitivity states, such as dynamic mechanical allodynia [34], mechanical pinprick hyperalgesia [35], and thermal allodynia [36] after capsaicin application. Moreover, two studies in human volunteers found that repeated sessions of rTMS over the left DLPFC can increase pressure pain thresholds and reduce muscle soreness, pain intensity, and size of the hyperalgesic area in an experimental model of muscle sustained pain (intramuscular injection of nerve growth factor) [12,13]. More recently, Vo *et al.* reported that anodal tDCS over the left DLPFC may modulate the intensity of secondary mechanical hyperalgesia induced by low-frequency electrical stimulation (LFS) [37].

It should be stressed that the effect of tDCS on the area of HFS-induced secondary hyperalgesia observed in the present study were modest as compared to the effects reported in those three studies [12,13,37]. Several factors could explain this, the first of which could be the montage that we used to target the left DLPFC. Traditionally, tDCS has been delivered through two large surface electrodes– one located over F3 and the other over the contralateral supraorbital region –inducing a very diffuse *E*-field [4]. This is the montage that was used in the aforementioned study by Vo *et al.* [37]. Here, we used a multichannel montage with smaller electrodes, with the aim of increasing the focality of stimulation and the selectivity of its neuromodulatory effects [6]. However, the superiority of multichannel montages in real-life stimulation remains to be demonstrated, especially regarding the DLPFC. While multichannel tDCS over the primary motor cortex (M1) increases motor cortex excitability more than conventional tDCS [38], a recent study investigating the effect of tDCS over the left DLPFC on working memory did not show an advantage of a multichannel montage over a traditional bipolar montage [39].

Second, our experimental protocol included only one 20-minute session of tDCS for each polarity. Stronger effects might have been observed after multiple sessions of tDCS. Indeed, it

has been demonstrated that sessions of tDCS over M1 repeated daily have a cumulative effect on motor cortex excitability [40]. Regarding the DLPFC, repeated sessions of rTMS led to a more pronounced reduction of hyperalgesia in an experimental model of muscle pain [12,13]. Of note, treatment of chronic pain with tDCS usually involves multiple sessions, while in the field of postoperative analgesia, only studies using several sessions over the first postoperative days reported positive results [2,18]. In contrast, a single 20-minutes anodal tDCS session did reduce the intensity of perception of LFS-induced hyperalgesia [37].

A fourth aspect to consider is the 30 minutes delay between the end of the tDCS session and the application of HFS. This interval was introduced in the study design because a previous study suggested that anodal tDCS targeting the left DLPFC increased M1 excitability and pain pressure threshold that tends to build up over time for at least such a delay [30]. However, other studies reported effects of tDCS over the DLPFC on pain perception or hyperalgesia either *during* stimulation or *just after* the end of stimulation [10,37,41,42]. Similarly, recent studies found a reduction of the intensity of mechanical secondary hyperalgesia induced by capsaicin when anodal tDCS over M1 was delivered either *during* or *after* capsaicin application [43,44].

Fifth, the left DLPFC might not be the most optimal target to modulate the state of descending modulation pathways. The most extensively studied cortical region with non-invasive neuromodulation techniques is M1, whose stimulation increases both pain thresholds and the effect of conditioned pain modulation in healthy volunteers [45]. Furthermore, using rTMS, modulation of M1 was more effective than modulation of the DLPFC at reducing heat pain intensity after capsaicin application [46]. Using tDCS, Meeker *et al.* demonstrated that one session of anodal stimulation targeting M1 – as compared to cathodal or sham stimulation– decreased the intensity of perception of mechanical pinprick stimuli after capsaicin-heat application [43]. Similarly, Hughes *et al.* showed that anodal tDCS over M1 reduces capsaicin-induced dynamic mechanical allodynia and mechanical pinprick hyperalgesia [44].

Finally, we must consider potential specificities of the different experimental procedures used to induce secondary hyperalgesia in healthy volunteers. As an example, NMDA antagonists attenuate mechanical hyperalgesia induced by capsaicin or thermal burns [47], but not by HFS [48]. While we have already cited several studies showing effects of non-invasive neuromodulation on secondary hyperalgesia induced by capsaicin [43,44] or LFS [37], very little data– besides those of the present study –are available regarding the effects of TMS or tDCS on HFS-induced hyperalgesia.

Although the effect was modest, our results do show an effect of anodal tDCS over the DLPFC on the spatial extent of HFS-induced secondary hyperalgesia. Therefore, the modest effect observed in our study should not necessarily preclude assessing the effects of tDCS over the DLPFC in patients exposed to the risk of developing secondary hyperalgesia, such as patients undergoing surgery. The programmed nature of the surgical insult provides us with a window of opportunity to implement primary preventive measures, aiming to interfere with the induction of postoperative secondary hyperalgesia [49]. The modulation of the extent of hyperalgesia could be particularly relevant in this population, as clinical studies have shown that the size of the area of postoperative secondary hyperalgesia is predictive for the development persistent pain after surgery [8–10]. Also, a recent meta-analysis found that the effect sizes of non-invasive stimulation of M1 on pain thresholds is twice as large in patients with pain as compared to healthy volunteers [44].

A first limitation our study is its relatively small sample size, which was based on the tested samples of previous studies investigating the effects of tDCS on nociceptive processing, the within-subject design of our study (which increases power), the necessity to use a multiple of six (to counterbalance the order of the sessions), and the desire to limit drop-outs (the protocol

required each subject to come back three times and be available for several hours for each session). Secondly, blinding may not have been optimal. The perception elicited by real tDCS is quite restricted to the beginning of the stimulation, which is well mimicked by the design of the sham stimulation. However, whether current sham tDCS protocols can ensure adequate blinding is debated, with recent studies reporting conflicting results [50–52]. To evaluate blinding in this study, we asked participants to guess if they received active or sham stimulation after each session and computed Cohen's κ to assess degree of agreement. Agreement was fair in the first session and poor in the second and third, overall indicating adequate blinding. Still, recent studies have suggested that an "end-of-study guess" might not be the best method to evaluate blinding during tDCS, especially at the current intensities used in the study [53]. Finally, a common limitation of within-subject designs is their susceptibility to carry-over and order effects. In this study, we tried to minimize both by separating the experimental sessions by a minimum of two weeks, by randomly assigning– in a counterbalanced way – the order of the treatment sessions, and by including session order as a random factor in our statistical model.

In conclusion, these results show that anodal multichannel tDCS targeting the left DLPFC– modestly –reduces the extent of HFS-induced secondary mechanical hyperalgesia. These results support the notion that non-invasive neuromodulation targeting the left DLPFC could be a potential intervention to prevent the development of central sensitization in patients.

## Acknowledgments

This research has benefited from the statistical advice of Cécile Bugli, Ph.D., of the Statistical Methodology and Computing Service, a technological platform at UCLouvain.

## Author Contributions

**Conceptualization:** Arnaud Steyaert, Cédric Lenoir, Patricia Lavand'homme, Emanuel N. van den Broeke, André Mouraux.

**Formal analysis:** Arnaud Steyaert.

**Funding acquisition:** Arnaud Steyaert, Patricia Lavand'homme, André Mouraux.

**Investigation:** Arnaud Steyaert.

**Methodology:** Arnaud Steyaert, Cédric Lenoir, Emanuel N. van den Broeke, André Mouraux.

**Project administration:** André Mouraux.

**Resources:** Arnaud Steyaert, Cédric Lenoir, André Mouraux.

**Supervision:** André Mouraux.

**Validation:** Arnaud Steyaert.

**Visualization:** Arnaud Steyaert.

**Writing – original draft:** Arnaud Steyaert.

**Writing – review & editing:** Arnaud Steyaert, Cédric Lenoir, Patricia Lavand'homme, Emanuel N. van den Broeke, André Mouraux.

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
