## [Decision Letter · Decision Letter 0]

24 Feb 2022

PONE-D-22-01850Multichannel transcranial direct current stimulation over the left dorsolateral prefrontal cortex may modulate the induction of secondary hyperalgesia, a double-blinded cross-over study in healthy volunteersPLOS ONE

Dear Dr. Steyaert,

Thank you for submitting your manuscript to PLOS ONE. After careful consideration, we feel that it has merit but does not fully meet PLOS ONE’s publication criteria as it currently stands. Therefore, we invite you to submit a revised version of the manuscript that addresses the points raised during the review process. First, I would like to apologize for the delay in providing a decision. I was waiting for a third opinion regarding your manuscript, but the appointed reviewer could not deliver in due time. At any rate, I have two reports to base my decision on. As you will see, Reviewer 2 has only some minor suggestions to improve the manuscript. However, Reviewer #1 has several questions regarding the methodological approach (e.g., the rationale for selecting the montage, potential issues with blinding, lack of a proper power analysis). I tend to agree with this Reviewer and invite you to provide a revised version addressing the comments raised by Reviewer #1 and those by Reviewer #2, as well (notably, the need to add a section the study limitations).

We look forward to receiving your revised manuscript.

Kind regards,

François Tremblay, PhD

Academic Editor

PLOS ONE

Journal Requirements:

[Declaration of interest: none. Arnaud Steyaert is supported for this research by a grant from the Fonds de Recherche Clinique (Cliniques Universitaires Saint-Luc, Belgium) and from the BESARPP (Belgian Society for Anaesthesiology and Reanimation). The funders had no role in study design, data collection and analysis, decision to publish, or preparation of the manu-script. This research has also benefited from the statistical advice of Cécile Bugli, Ph.D., of the Statistical Methodology and Computing Service, a technological platform at UCLouvain.]

 [AS is supported for this research by a grant from the Fonds de Recherche Clinique (Cliniques Universitaires Saint-Luc, Belgium) and from the BESARPP (Belgian Society for Anaesthesiology and Reanimation). The funders had no role in study design, data collection and analysis, decision to publish, or preparation of the manuscript.]

Reviewers' comments:

Reviewer's Responses to Questions

**Comments to the Author**

1. Is the manuscript technically sound, and do the data support the conclusions?

Reviewer #1: Yes

Reviewer #2: Yes

2. Has the statistical analysis been performed appropriately and rigorously? 

Reviewer #1: Yes

Reviewer #2: Yes

3. Have the authors made all data underlying the findings in their manuscript fully available?

Reviewer #1: Yes

Reviewer #2: Yes

4. Is the manuscript presented in an intelligible fashion and written in standard English?

Reviewer #1: Yes

Reviewer #2: Yes

5. Review Comments to the Author

Reviewer #1: It is a well-designed study that investigated DLPFC-tDCS effects on the secondary mechanical hyperalgesia. Results showed that anodal tDCS reduced the area of hyperalgesia. It suggests that left DLPFC stimulation may be used to prevent the development of central sensitization. I have some suggestions that hope the authors the address.

(1) While the target region is left DLPFC, this study used a multichannel tDCS with different current weight on different electrodes. It is also different from the montage of HD-tDCS. I don’t understand the rational of using this stimulation montage, particularly how the authors define the current weight on the 5 electrodes. I suggest the authors can provide some evidence or references on using this montage, and the comparisons between this montage and traditional montage are preferred.

(2) Immediately before and after tDCS, the detection threshold to a single electrical pulse was measured. What is the aim for this sensory test? In addition, how about tDCS effects on the detection threshold to the electrical stimulus? I suggest the authors to report these results.

(3) Although tDCS did not significantly alter intensity perception to mechanical pinprick stimuli at the HFS arm, the area of hyperalgesia was reduced after anodal tDCS. I am wondering about the relationship between tDCS effects on reducing hyperalgesia area and those on altering secondary hyperalgeisa effect. I would suggest perform a correlation analysis to clarify this issue.

(4) What is the rational for the sample size? Any power analysis was performed prior the study? The comparable studies on the 100th line should be clearly cited in the manuscript.

(5) The blinding effect is poor. Is it likely that the current intensity was 2 mA and the perception is quite different between active and sham condition? This is a limitation of this study and should be discussed.

(6) This study provided evidence for the effects of tDCS on the extent of the area of HFS-induced secondary hyperalgesia. Some studies have shown DLPFC-tDCS can alleviate the affective response to pain, but why anodal tDCS over the left DLPFC can be used to reduce hyperalgesia. I would suggest the authors can discuss the underlying neural mechanisms. In addition, why the left DLPFC is targeted, instead of the right side?

Reviewer #2: Thank you for the opportunity to read this article. I find it very well done, well written and methodologically very rigorous.

I might suggest that the authors add what the manuscript's limitations are.

Furthermore, I would suggest to better highlight the reasons that led to having such a small sample of numbers (for example, the complex methodology that requires several observations and a considerable commitment on the part of the subjects).

A final suggestion is to make more explicit what the contribution of this study may be to clinical practice (for example, recent manuscript 10.12788 / acp.0009).

6. PLOS authors have the option to publish the peer review history of their article (what does this mean?). If published, this will include your full peer review and any attached files.

Reviewer #1: No

Reviewer #2: No

---

## [Author Response · Author response to Decision Letter 0]

9 May 2022

Editorial requests

(1) The manuscript has been reformatted to meet PLOS ONE's style requirements.

(2) As requested, we have removed the following funding information from the manuscript: AS is supported for this research by a grant from the Fonds de Recherche Clinique (Cliniques Universitaires Saint-Luc, Belgium) and from the BESARPP (Belgian Society for Anaesthesiology and Reanimation). The funders had no role in study design, data collection and analysis, decision to publish, or preparation of the manuscript.

(3) We are working on the preparation of the data file for the repository and will be able to provide a DOI rapidly if the article is accepted.

Point-to-point response to reviewers (a formatted version of these responses can be found in the pdf of the submission)

Reviewer #1

It is a well-designed study that investigated DLPFC-tDCS effects on the secondary mechanical hyperalgesia. Results showed that anodal tDCS reduced the area of hyperalgesia. It suggests that left DLPFC stimulation may be used to prevent the development of central sensitization. I have some suggestions that I hope the authors the address. 

We thank the reviewer for his positive and constructive suggestions, which we address in the following paragraphs.

(1) While the target region is left DLPFC, this study used a multi-channel tDCS with different current weight on different electrodes. It is also different from the montage of HD-tDCS. I don’t understand the rational of using this stimulation montage, particularly how the authors define the current weight on the 5 electrodes. I suggest the authors can provide some evidence or references on using this montage, and the comparisons between this montage and traditional montage are preferred. 

Our goal with the use of this montage – rather than a more traditional bipolar one – was to improve the spatial focality – and therefore the specificity – of stimulation. We relied on a custom 5-electrode montage recommended by the manufacturer for use with the Starstim current generator and based on the modelling work by Ruffini et al. [1]. The overall aim is to maximize the normal E-field (En) on target, while minimizing the En elsewhere. After defining the target area (left DLPFC or BA46), the En in the target area (0.25 V/m), the size and number of electrodes (3.14 cm² and 5), and the maximum current per electrode (2 mA), the optimal multichannel montage was determined by minimizing the least-square difference between the resulting E-field and a weighted target map of the En.

While multichannel montages are theoretically able to improve both focality of stimulation and average En on target, their superiority over bipolar montage remains to be demonstrated in real-life stimulation. At the time of study inception, there were – to the best of our knowledge – no studies available comparing such a montage to a bipolar one. Since then, however, a study investigating the effect of tDCS over the left DLPFC on working memory did not show an advantage of a multichannel montage over a traditional bipolar montage [2]. This has been mentioned in the discussion. Finally, it should be noted that the the optimisation of the montages is still a quite recent technique and that new algorithms are currently under development [3]. 

(2) Immediately before and after tDCS, the detection threshold to a single electrical pulse was measured. What is the aim for this sensory test? In addition, how about tDCS effects on the detection threshold to the electrical stimulus? I suggest the authors to report these results.

We added this additional testing point to verify that the tDCS session did not modify the electrical detection threshold, and therefore the ability of HFS to activate primary afferents. To analyze that, we fitted a linear mixed model on the square root of the electrical detection threshold with as fixed effects “time” (baseline and T–5), “stimulation” (anodal, cathodal, and sham), and their interactions. We included a random intercept for “participants” and “session”. We found no effect of time (F (1, 83) = 1.36, p = 0.25), of stimulation (F (2, 83) = 1,74, p = 0.18) or of their interaction (F (2, 83) = 1.37, p = 0.26). We have reported these results in the paper.

(3) Although tDCS did not significantly alter intensity perception to mechanical pinprick stimuli at the HFS arm, the area of hyperalgesia was reduced after anodal tDCS. I am wondering about the relationship between tDCS effects on reducing hyperalgesia area and those on altering secondary hyperalgesia effect. I would suggest performing a correlation analysis to clarify this issue. 

The suggested correlation analysis showed that overall, the percentage of change in pinprick sensitivity after HFS was significantly correlated with the extent of the area of secondary hyperalgesia (ρ = 0.35, p < 0.001). Sub-group analysis according to tDCS condition showed that this correlation remained statistically significant after sham (ρ = 0.40, p = 0.002) and cathodal (ρ = 0.40, p < 0.002), but not after anodal stimulation (ρ = 0.22, p = 0.10). While this could suggest that intensity of perception and area of hyperalgesia were indeed modulated differently according to the type of stimulation, these differences in correlation across tDCS conditions were not statistically significant (z = 0.723, p = 0.22). We do not feel that reporting these unplanned analyses would really add to the manuscript.

(4) What is the rational for the sample size? Any power analysis was performed prior the study? The comparable studies on the 100th line should be clearly cited in the manuscript. 

We based our sample size on (1) the tested samples of previous studies investigating the effects of tDCS on nociceptive processing (see [4] for a systematic review), (2) the within-subject design of our study, (3) the necessity to use a multiple of six (to counter-balance the order of the sessions), and (4) the desire to limit drop-outs (as the protocol required each subject to come back three times and be available for several hours for each session).

(5) The blinding effect is poor. Is it likely that the current intensity was 2 mA, and the perception is quite different between active and sham condition? This is a limitation of this study and should be discussed. 

The perception elicited by real tDCS is quite restricted to the beginning of the stimulation, which is well mimicked by the design of the sham stimulation. However, whether current sham tDCS protocols can ensure adequate blinding is debated, with recent studies reporting conflicting results [5–7]. To evaluate blinding in this study, we asked the participants to guess if they received active or sham stimulation after each session and computed Cohen’s κ to assess degree of agreement. While agreement was fair in the first session (κ = 0.30, 95%IC –0.13 to 0.74), it was poor in the second κ = –0.17, 95%IC –0.59 to 0.24) and the third κ = –0.09, 95%IC –0.51 to 0.33), overall indicating adequate blinding. Still, we acknowledge that the “end-of-study guess” might not have been the best method of evaluating blinding during tDCS, especially at the current intensities used in the study [8] and have added this to the limitations.

(6.1) This study provided evidence for the effects of tDCS on the extent of the area of HFS-induced secondary hyperalgesia. Some studies have shown DLPFC-tDCS can alleviate the affective response to pain, but why anodal tDCS over the left DLPFC can be used to reduce hyperalgesia. I would suggest the authors can discuss the underlying neural mechanisms.

Neuromodulation of the DLPFC could affect responses to painful stimuli and the development of hyperalgesia through similar or different mechanisms. Secondary hyperalgesia has been explained as resulting from an enhanced synaptic transmission of nociceptive input at the level of the spinal cord [9]. Importantly, nociceptive circuits at spinal level can also be facilitated or inhibited by descending modulation pathways originating from supra-spinal structures, e.g., the rostral ventral medulla and the periaqueductal gray [10]. An MRI-study using tensor diffusion imaging techniques has demonstrated anatomical connectivities between these structures and the DLPFC [11], and it has been suggested that the DLPFC could exert inhibition of the midbrain-thalamic pathway [12]. More recently, animal work by Tan et al. demonstrated that the midcingulate cortex (MCC) is crucial in regulating mechanical hypersensitivity behaviour after intraplantar capsaicin injection in mice [13]. While it is difficult to target this deep brain structure using current non-invasive neuromodulation techniques, an indirect modulation of the MCC could be achieved by targeting the left DLPFC. Indeed, Stagg et al. [12] showed that anodal tDCS over the left DLPFC induces widespread changes in brain perfusion, including in the thalamus and the MCC [25].

(6.2) In addition, why the left DLPFC is targeted, instead of the right side?

The differential role of the left and right DLPFC in the perception of nociceptive stimuli has not been demonstrated conclusively [14]. Both seem to be activated following nociceptive stimulation and structural modifications have been reported for both sides in chronic pain patients [14]. We targeted the left DLPFC because: (1) it has been shown that its modulation induces bilateral antinociceptive effects [15], (2) its modulation reduces hyperalgesia in a model of sustained muscle pain [16], and that (3) most clinical studies use the same target, increasing translatability [17–19].

Reviewer #2

Thank you for the opportunity to read this article. I find it very well done, well written and methodologically very rigorous. 

We thank the reviewer for his nice remarks and address his suggestions in the following paragraphs.

(1) I might suggest that the authors add what the manuscript's limitations are. 

We totally agree with the reviewer and have now expanded and clarified the last paragraph of the discussion, to include and summarize the main limitations of the study, including the small sample size (see below).

(2) Furthermore, I would suggest to better highlight the reasons that led to having such a small sample of numbers (for example, the complex methodology that requires several observations and a considerable commitment on the part of the subjects). 

Indeed, the concern for dropouts was one of the factors we considered to set our sample size. As mentioned in the reply to point (4) of Reviewer #1, the sample size was based on (1) the samples of previous studies investigating the effects of tDCS on nociceptive processing (see [4] for a systematic review), (2) the within-subject design of our study, (3) the necessity to use a multiple of six (to counterbalance the order of the sessions), and (4) the desire to limit drop-outs (as the protocol required each subject to come back three times and be available for several hours for each session). We have added this to the discussion of limitations.

(3) A final suggestion is to make more explicit what the contribution of this study may be to clinical practice (for example, recent manuscript 10.12788 / acp.0009). 

We thank the reviewer for this relevant suggestion. We have expanded the next-to-last paragraph of the discussion to better emphasize the potential contribution of our findings to clinical practice. The effect on the extent of hyperalgesia could be particularly relevant, as clinical studies have shown that the size of the area of postoperative secondary hyperalgesia is predictive for the development persistent pain after surgery [20–22]. The programmed nature of the surgical insult provides us with a window of opportunity to implement primary preventive measures, aiming to interfere with the induction of postoperative secondary hyperalgesia [23].

References

1. Ruffini G, Fox MD, Ripolles O, Miranda PC, Pascual-Leone A. Optimization of multifocal transcranial current stimulation for weighted cortical pattern targeting from realistic modeling of electric fields. Neuroimage. 2014;89: 216–225. doi:10.1016/j.neuroimage.2013.12.002

2. Splittgerber M, Salvador R, Brauer H, Breitling-Ziegler C, Prehn-Kristensen A, Krauel K, et al. Individual Baseline Performance and Electrode Montage Impact on the Effects of Anodal tDCS Over the Left Dorsolateral Prefrontal Cortex. Front Hum Neurosci. 2020;14: 349. doi:10.3389/fnhum.2020.00349

3. Saturnino GB, Madsen KH, Thielscher A. Optimizing the electric field strength in multiple targets for multichannel transcranial electric stimulation. J Neural Eng. 2021;18: 014001. doi:10.1088/1741-2552/abca15

4. Giannoni-Luza S, Pacheco-Barrios K, Cardenas-Rojas A, Mejia-Pando PF, Luna-Cuadros MA, Barouh JL, et al. Noninvasive motor cortex stimulation effects on quantitative sensory testing in healthy and chronic pain subjects: a systematic review and meta-analysis. Pain. 2020;161: 1955–1975. doi:10.1097/j.pain.0000000000001893

5. O’Connell NE, Cossar J, Marston L, Wand BM, Bunce D, Moseley GL, et al. Rethinking Clini-cal Trials of Transcranial Direct Current Stimulation: Participant and Assessor Blinding Is Inade-quate at Intensities of 2mA. Plos One. 2012;7: e47514. doi:10.1371/journal.pone.0047514

6. Dinn W, Göral F, Adigüzel S, Karamürsel S, Fregni F, Aycicegi-Dinn A. Effectiveness of tDCS blinding protocol in a sham-controlled study. Brain Stimul. 2017;10: 401. doi:10.1016/j.brs.2017.01.188

7. Reckow J, Rahman-Filipiak A, Garcia S, Schlaefflin S, Calhoun O, DaSilva AF, et al. Tolerabil-ity and blinding of 4x1 high-definition transcranial direct current stimulation (HD-tDCS) at two and three milliamps. Brain Stimul. 2018;11: 991–997. doi:10.1016/j.brs.2018.04.022

8. Turner C, Jackson C, Learmonth G. Is the “end‐of‐study guess” a valid measure of sham blinding during transcranial direct current stimulation? European J Neurosci. 2021;53: 1592–1604. doi:10.1111/ejn.15018

9. Woolf CJ. Central sensitization: implications for the diagnosis and treatment of pain. Pain. 2011;152: S2 15. doi:10.1016/j.pain.2010.09.030

10. Sandkühler J. Models and Mechanisms of Hyperalgesia and Allodynia. Physiol Rev. 2009;89: 707–758. doi:10.1152/physrev.00025.2008

11. Hadjipavlou G, Dunckley P, Behrens TE, Tracey I. Determining anatomical connectivities between cortical and brainstem pain processing regions in humans: A diffusion tensor imaging study in healthy controls. Pain. 2006;123: 169–178. doi:10.1016/j.pain.2006.02.027

12. Stagg CJ, Lin RL, Mezue M, Segerdahl A, Kong Y, Xie J, et al. Widespread Modulation of Cerebral Perfusion Induced during and after Transcranial Direct Current Stimulation Applied to the Left Dorsolateral Prefrontal Cortex. J Neurosci. 2013;33: 11425 11431. doi:10.1523/jneurosci.3887-12.2013

13. Tan LL, Pelzer P, Heinl C, Tang W, Gangadharan V, Flor H, et al. A pathway from midcingu-late cortex to posterior insula gates nociceptive hypersensitivity. Nat Neurosci. 2017;20: 1591 1601. doi:10.1038/nn.4645

14. Seminowicz DA, Moayedi M. The Dorsolateral Prefrontal Cortex in Acute and Chronic Pain. J Pain. 2017;18: 1027 1035. doi:10.1016/j.jpain.2017.03.008

15. Brighina F, Tommaso MD, Giglia F, Scalia S, Cosentino G, Puma A, et al. Modulation of pain perception by transcranial magnetic stimulation of left prefrontal cortex. J Headache Pain. 2011;12: 185–191. doi:10.1007/s10194-011-0322-8

16. Seminowicz DA, Martino E de, Schabrun SM, Graven-Nielsen T. Left dorsolateral prefrontal cortex repetitive transcranial magnetic stimulation reduces the development of long-term muscle pain. Pain. 2018;159: 2486–2492. doi:10.1097/j.pain.0000000000001350

17. O’Connell NE, Marston L, Spencer S, DeSouza LH, Wand BM. Non-invasive brain stimula-tion techniques for chronic pain. Pain C, Palliative, Group SC, editors. Cochrane Db Syst Rev. 2018;4: CD008208. doi:10.1002/14651858.cd008208.pub5

18. Steyaert A, Lenoir C, Lavand’homme P, Mouraux A. Transcranial direct current stimula-tion as a tool for postoperative pain management : a review of the current clinical evidence. Acta Anaesth Belg. 2019;7: 175–183.

19. Che X, Cash RFH, Luo X, Luo H, Lu X, Xu F, et al. High-frequency rTMS over the dorsolat-eral prefrontal cortex on chronic and provoked pain: A systematic review and meta-analysis. Brain Stimul. 2021;14: 1135–1146. doi:10.1016/j.brs.2021.07.004

20. Kock MD, Lavand’homme P, Waterloos H. “Balanced analgesia” in the perioperative period: is there a place for ketamine? Pain. 2001;92: 373 380.

21. Lavand’homme P, Kock MD, Waterloos H. Intraoperative epidural analgesia combined with ketamine provides effective preventive analgesia in patients undergoing major digestive surgery. Anesthesiology. 2005;103: 813 820.

22. Kock MD, Lavand’homme P, Waterloos H. The Short-Lasting Analgesia and Long-Term Antihyperalgesic Effect of Intrathecal Clonidine in Patients Undergoing Colonic Surgery. Anes-thesia Analgesia. 2005;101: 566–572. doi:10.1213/01.ane.0000157121.71808.04

23. Deumens R, Steyaert A, Forget P, Schubert M, Lavand’homme P, Hermans E, et al. Preven-tion of chronic postoperative pain: Cellular, molecular, and clinical insights for mechanism-based treatment approaches. Prog Neurobiol. 2013;104: 1–37. doi:10.1016/j.pneurobio.2013.01.002

---

## [Decision Letter · Decision Letter 1]

3 Jun 2022

Multichannel transcranial direct current stimulation over the left dorsolateral prefrontal cortex may modulate the induction of secondary hyperalgesia, a double-blinded cross-over study in healthy volunteers

PONE-D-22-01850R1

Dear Dr. Steyaert,

We’re pleased to inform you that your manuscript has been judged scientifically suitable for publication and will be formally accepted for publication once it meets all outstanding technical requirements.

Kind regards,

François Tremblay, PhD

Academic Editor

PLOS ONE

Additional Editor Comments (optional):

Reviewers' comments:

Reviewer's Responses to Questions

**Comments to the Author**

1. If the authors have adequately addressed your comments raised in a previous round of review and you feel that this manuscript is now acceptable for publication, you may indicate that here to bypass the “Comments to the Author” section, enter your conflict of interest statement in the “Confidential to Editor” section, and submit your "Accept" recommendation.

Reviewer #1: All comments have been addressed

2. Is the manuscript technically sound, and do the data support the conclusions?

Reviewer #1: Yes

3. Has the statistical analysis been performed appropriately and rigorously? 

Reviewer #1: Yes

4. Have the authors made all data underlying the findings in their manuscript fully available?

Reviewer #1: (No Response)

5. Is the manuscript presented in an intelligible fashion and written in standard English?

Reviewer #1: Yes

6. Review Comments to the Author

Reviewer #1: (No Response)

7. PLOS authors have the option to publish the peer review history of their article (what does this mean?). If published, this will include your full peer review and any attached files.

Reviewer #1: No

---

## [Editor Report · Acceptance letter]

8 Jun 2022

PONE-D-22-01850R1 

Multichannel transcranial direct current stimulation over the left dorsolateral prefrontal cortex may modulate the induction of secondary hyperalgesia, a double-blinded cross-over study in healthy volunteers 

Dear Dr. Steyaert:

I'm pleased to inform you that your manuscript has been deemed suitable for publication in PLOS ONE. Congratulations! Your manuscript is now with our production department. 

Kind regards, 

on behalf of

Dr. François Tremblay 

Academic Editor

PLOS ONE